# A Comprehensive Review of Antiviral Therapy for Hepatitis C: The Long Journey from Interferon to Pan-Genotypic Direct-Acting Antivirals (DAAs)

**DOI:** 10.3390/v17020163

**Published:** 2025-01-24

**Authors:** Lorenza Di Marco, Simona Cannova, Emanuele Ferrigno, Giuseppe Landro, Rosario Nonni, Claudia La Mantia, Fabio Cartabellotta, Vincenza Calvaruso, Vito Di Marco

**Affiliations:** 1SIcilian Network for Therapy, Epidemiology and Screening In Hepatology (SINTESI), 90127 Palermo, Italy; lorenza.dimarco@unimore.it (L.D.M.); cartabellotta.fabio@fbfpa.it (F.C.); vincenza.calvaruso@unipa.it (V.C.); 2Department of Oncology and Hematology, Azienda Ospedaliero-University Hospital of Mod, 41124 Modena, Italy; 3Clinical and Experimental Medicine PhD Program, University of Modena and Reggio Emilia, 41100 Modena, Italy; 4Department of Health Promotion, Mother and Child Care, Internal Medicine and Medical Specialties (PROMISE), University of Palermo, 90127 Palermo, Italy; simicannova@hotmail.it (S.C.); e.ferrigno14@gmail.com (E.F.); peppe.landro@virgilio.it (G.L.); rosariononni@hotmail.it (R.N.); claudialamantia1990@gmail.com (C.L.M.); 5Department of Medicine, Buccheri-La Ferla Hospital, 90123 Palermo, Italy

**Keywords:** pan-genotypic, direct-acting antivirals DAAs, HCV, HCV therapy, peg-IFN, sustained virological response, SVR

## Abstract

The treatment landscape for hepatitis C virus (HCV) infection has transformed over the past few decades, evolving from the limited efficacy of interferon (IFN) monotherapy to the highly successful pan-genotypic direct-acting antivirals (DAAs) used today. Initially, alpha-interferon monotherapy, introduced in the 1990s, was the standard treatment, yet it provided low sustained virological response (SVR) rates and caused significant adverse effects, limiting its utility. The development of pegylated interferon (peg-IFN) improved the pharmacokinetic profile of IFN, allowing for less frequent dosing and modestly improved response rates. When combined with ribavirin, peg-IFN achieved higher SVR rates, especially in non-genotype 1 HCV infections, but the combination also brought additional side effects, such as anemia and depression. The advent of the first-generation DAAs, such as telaprevir and boceprevir, marked a significant milestone. Combined with peg-IFN and ribavirin, these protease inhibitors boosted response rates in patients with genotype 1 HCV. However, high rates of adverse effects and drug resistance remained challenges. Second-generation DAAs, like sofosbuvir and ledipasvir, introduced IFN-free regimens with improved safety profiles and efficacy. The most recent advances are pan-genotypic DAAs, including glecaprevir-pibrentasvir and sofosbuvir-velpatasvir, which offer high SVR rates across all genotypes, shorter treatment durations, and fewer side effects. Current pan-genotypic regimens represent a cornerstone in HCV therapy, providing an accessible and effective solution globally.

## 1. Introduction

The World Health Organization (WHO) estimates that 50 million people with chronic hepatitis C virus (HCV) infection live in the world, and 1 million new infections occur annually, with a global prevalence of approximately 1%. HCV is a leading cause of chronic liver disease, including cirrhosis and hepatocellular carcinoma (HCC) [1,2].

Chronic HCV infection is often asymptomatic, progressing over decades without detection. Without intervention, approximately 20–30% of chronically infected individuals develop liver cirrhosis, and up to 5% may develop HCC [3,4,5].

This progression highlights the need for effective antiviral therapy to prevent long-term liver damage and associated mortality. Antiviral therapy has been shown to reduce the incidence of cirrhosis, liver failure, and HCC by achieving sustained virological response (SVR), essentially curing the infection [6,7].

The development of antiviral therapy for HCV has been transformative. Treatment has evolved from interferon-based regimens, which were only modestly effective and associated with severe side effects, to highly effective direct-acting antivirals (DAAs) that provide high SVR rates with minimal adverse effects [8]. The latest pan-genotypic DAAs, effective across all HCV genotypes, have simplified treatment protocols, making them accessible and effective worldwide [9,10].

This review aims to provide a comprehensive overview of the evolution of antiviral therapy for HCV, tracing the journey from early interferon-based regimens to the development of pan-genotypic DAAs. By examining each treatment stage’s efficacy, safety, and limitations, it seeks to highlight the pivotal advances in HCV therapy and discuss the implications for global health and HCV elimination efforts.

The main purpose of the review is to describe the progressive and rapid effectiveness of drugs for all patients with hepatitis C, a disease for which, until 35 years ago, the etiological agent was unknown and had no cure. In light of these powerful innovations, DAAs should be available not only for developed countries, but also and at less onerous costs, for developing countries where the prevalence of HCV infection and liver disease is very high.

## 2. The Era of Alpha-Interferon Monotherapy

In the early 1990s, alpha-interferon (IFN) monotherapy emerged as the first-line treatment for HCV infection, marking a milestone in the management of a disease that was often asymptomatic but could progress to severe liver damage over time [11].

## 3. Mechanism of Action of Alpha-Interferon

IFN is a cytokine produced by immune cells in response to viral infections. It functions by inducing an “antiviral state” within host cells. The antiviral effects are mediated through several pathways: alpha-interferon binds to receptors on the surface of target cells, activating intracellular signaling pathways, such as the JAK-STAT pathway, which leads to the transcription of interferon-stimulated genes (ISGs). These genes produce antiviral proteins that inhibit viral replication at various stages, including entry, RNA transcription, protein synthesis, and release. Interferon also enhances the immune response by promoting the activity of natural killer (NK) cells and cytotoxic T lymphocytes, which play critical roles in clearing infected cells [12].

## 4. Clinical Efficacy and Challenges

While alpha-interferon represented a significant advancement, the therapy was limited by low response rates and high rates of relapse post-treatment. Early studies showed that ALT levels were normalized in 35–40% of patients, and biopsy also demonstrated histological improvement in lobular and periportal inflammation. Unfortunately, relapse was a common problem in these studies, occurring in 49–57% of patients [13,14,15].

The poor response rates were compounded by the significant side effects associated with interferon therapy. Common adverse effects included flu-like symptoms (fever, fatigue, myalgia), neuropsychiatric symptoms (depression, irritability), and hematologic changes, such as leukopenia and thrombocytopenia, which often required dose adjustments or even discontinuation of therapy [16].

## 5. Understanding the Limitations

Research during this era highlighted the inherent limitations of alpha-interferon in treating HCV. Unlike other viruses that rely on intracellular replication machinery (such as the hepatitis B virus), HCV has a high mutation rate, resulting in quasi-species or closely related viral variants that can evade immune responses. This genetic variability made HCV particularly resistant to interferon monotherapy, which could not completely suppress viral replication [17,18,19].

Additionally, the need for frequent subcutaneous administration, often three times weekly, the limited efficacy, and the high side effect rates of alpha-interferon monotherapy underscored the need for other treatments that could enhance effectiveness and safety.

## 6. Pegylated Interferon Monotherapy

The introduction of pegylated interferon (peg-IFN) in the early 2000s represented a pivotal advancement in HCV treatment. Chemically modifying IFN with polyethylene glycol (PEG) offered a more stable pharmacokinetic profile, allowing for once-weekly dosing while enhancing efficacy in suppressing HCV replication [20].

## 7. Mechanism and Pharmacokinetics of Pegylation

Pegylation is a process that involves attaching PEG, a large, inert molecule, to the interferon protein. This modification shields the interferon from rapid degradation and clearance, particularly by the kidneys, and reduces its immunogenicity, thereby prolonging its half-life. The PEG molecule increases the interferon’s size, slowing its absorption and creating a controlled release mechanism that maintains stable blood levels over the week. The two primary peg-IFN formulations differ slightly in their PEG structure: peg-IFN alfa-2a uses a larger, branched PEG molecule, which provides more consistent plasma concentrations, while peg-IFN alfa-2b uses a smaller, linear PEG molecule, which is absorbed more rapidly but still offers a significant half-life extension compared to standard IFN [21]. This prolonged and stable pharmacokinetic profile was beneficial for HCV treatment, as it maintained adequate drug levels without the peaks and troughs associated with standard IFN. This not only provided continuous antiviral action but also minimized the frequency of side effects that tended to arise during the peaks in standard IFN dosing [22].

## 8. Challenges and Limitations of Peg-IFN Monotherapy

While peg-IFN significantly improved over standard IFN, its efficacy as a monotherapy was limited. SVR rates with peg-IFN monotherapy remained suboptimal, particularly in genotype 1 patients, with approximately 60–70% of genotype 1 patients experiencing viral relapse after treatment [23,24].

## 9. Clinical Implications of Peg-IFN Monotherapy

Despite its limitations, peg-IFN monotherapy marked a significant step forward in HCV treatment and set the stage for combination therapy with ribavirin. The enhanced pharmacokinetics and improved SVR rates of peg-IFN compared to standard IFN provided the foundation for subsequent treatment advancements. Peg-IFN also highlighted the need for genotype-specific approaches, as genotypes 1 and 4 proved more challenging to treat with peg-IFN monotherapy [25,26], while genotypes 2 and 3 were more responsive [27]. These findings underscored the genetic diversity of HCV and informed subsequent treatment guidelines and strategies [28,29].

## 10. Combination Therapy with Pegylated Interferon and Ribavirin

The combination of peg-IFN with ribavirin marked a significant milestone in HCV treatment, substantially improving SVR rates compared to peg-IFN alone. Introduced in the early 2000s, this combination became the standard of care and remained the cornerstone of HCV therapy for over a decade until the advent of DAAs. Ribavirin, a nucleoside analog with broad-spectrum antiviral activity, enhanced the effectiveness of peg-IFN, particularly in achieving SVR across various HCV genotypes, though challenges persisted, especially for genotypes 1 and 4 [30].

## 11. Mechanism of Action of Ribavirin and Its Role in Improving SVR

Ribavirin’s mechanism of action is complex. It is thought to exert its antiviral effects through multiple pathways, including (1) the direct inhibition of HCV replication, (2) inhibition of host inosine monophosphate dehydrogenase (IMPDH) enzyme, (3) mutagenesis induction to drive a rapidly replicating virus beyond the threshold to error catastrophe, and (4) immunomodulation by inducing a Th1 immune response. Specifically, ribavirin monophosphate (RMP) is converted to ribavirin triphosphate (RTP) forms, and binding of RTP to the nucleotide binding site of RNA polymerase prevents binding of the correct nucleotides, leading to reduced viral replication or the production of defective virions. Additionally, ribavirin monophosphate is a competitive inhibitor of IMPDH, an enzyme involved in the de novo synthesis of guanine nucleotides. This could lead to a decline in viral protein synthesis and limit the replication of viral genomes. Finally, ribavirin enhances the immune response against HCV by modulating the balance of T-helper (Th1/Th2) cells, favoring a Th1-dominant, cell-mediated immune response, which is more effective in clearing the virus [31].

In combination with peg-IFN, ribavirin enhances the SVR by increasing the likelihood of viral eradication, particularly when administered over an extended period [32].

## 12. SVR Rates Across Different HCV Genotypes

HCV genotype plays a crucial role in determining treatment response. Peg-IFN alfa-2b plus ribavirin achieved significantly higher SVR rates compared to interferon alfa-2b plus ribavirin across all HCV genotypes (Table 1). However, response rates varied by genotype. For patients with genotype 1, the most prevalent and hardest-to-treat genotype, the SVR rate was around 42% with combination therapy, compared to lower rates with interferon alone. This represented a meaningful improvement but left a substantial proportion of genotype 1 patients without an effective cure.

In contrast, patients with genotypes 2 and 3 responded much more favorably to the combination therapy, with SVR rates of around 76–82%. This high efficacy in genotypes 2 and 3 led to shorter treatment durations (24 weeks instead of 48 weeks), given that these genotypes were more susceptible to the antiviral effects of the combined therapy. Genotype 4, more prevalent in regions such as the Middle East and North Africa, exhibited intermediate SVR rates of approximately 55–60%, presenting additional challenges in achieving complete viral eradication compared to genotypes 2 and 3 [33,34,35,36,37,38,39,40,41].

## 13. Adverse Effects of Ribavirin, Especially Hematological Toxicity

While ribavirin enhanced treatment efficacy, it also introduced significant adverse effects, particularly hematological toxicity. The most notable side effect is hemolytic anemia, which occurs because ribavirin accumulates in red blood cells, causing oxidative damage that leads to hemolysis [42]. Ribavirin can also exacerbate the neuropsychiatric side effects associated with interferon, such as depression and irritability, impacting patient adherence [43,44].

## 14. The Introduction of Direct-Acting Antivirals (DAAs): First Generation

The introduction of DAAs marked a turning point in HCV treatment, revolutionizing approaches with enhanced efficacy compared to prior therapies. The first-generation DAAs, notably protease inhibitors such as boceprevir and telaprevir, were designed to target viral replication, significantly advancing the treatment landscape for chronic HCV patients.

## 15. Mechanism of Action: NS3/4A Protease Inhibition

Boceprevir and telaprevir became FDA-approved in 2011 as the first DAAs targeting the HCV NS3/4A protease, a crucial enzyme in viral replication [45,46]. By binding to the active site of this protease, boceprevir and telaprevir effectively turn off the protease’s function, halting the replication process. Unlike previous interferon-based therapies, these protease inhibitors offered a mechanism of action specific to the virus, opening the door to targeted antiviral therapy and transforming the treatment paradigm by focusing on viral pathways rather than general immune system modulation [47,48,49].

## 16. Improvements in Sustained Virological Response (SVR) Rates

One of the key benefits of first-generation DAAs was their impact on SVR rates, a crucial indicator of HCV treatment efficacy. Clinical trials demonstrated that adding boceprevir or telaprevir to the existing standard of care—typically PEG-IFN and ribavirin—dramatically improved SVR rates, especially in patients with genotype 1 HCV, the most challenging to treat with IFN-based therapies alone (Table 2). For example, telaprevir-based regimens achieved SVR rates between 67% and 79% in treatment-naïve genotype 1 patients, a substantial improvement over the approximate 40% SVR rates seen with PEG-IFN and ribavirin alone [50,51,52,53,54,55,56,57].

## 17. Challenges and Limitations: Resistance, Complex Dosing, and Drug–Drug Interactions

Despite their efficacy, first-generation DAAs presented significant challenges, including the risk of viral resistance, adverse effects, and complex treatment regimens. Resistance was a notable issue; since these DAAs targeted a single viral enzyme, mutations in the NS3/4A protease could render the drugs less effective. Resistance-associated variants (RAVs) often emerged during treatment, particularly in patients who experienced incomplete viral suppression. This challenge emphasized the need for combination therapies targeting multiple viral components to reduce the risk of resistance [58].

Another major limitation of first-generation DAAs was their reliance on interferon. Boceprevir and telaprevir were not administered as standalone therapies; rather, they were combined with PEG-IFN and ribavirin, which brought additional adverse effects and lengthy treatment durations. IFN-based regimens were associated with side effects such as fatigue, flu-like symptoms, and hematologic abnormalities, often making adherence challenging for patients. Moreover, the DAAs themselves introduced side effects, including anemia, rash, and gastrointestinal disturbances, further complicating treatment [59,60,61].

Finally, first-generation DAAs also required intricate dosing schedules, demanding strict patient adherence. The potential for drug–drug interactions exacerbated this complexity, as boceprevir and telaprevir were metabolized by the liver enzyme CYP3A. This led to interactions with many other medications, particularly those affecting hepatic function. The narrow therapeutic window of these drugs made managing comorbid conditions challenging and sometimes necessitated additional monitoring or dose adjustments for concomitant medications [62,63,64,65].


**Biological insights that led to development of DAAs**


The development of DAAs was driven by several pivotal biological insights into the virus’s life cycle and molecular biology. These insights enabled the identification of highly specific therapeutic targets within the virus (Table 3).

Understanding the HCV Genome and Polyprotein Processing

The HCV genome is a positive-sense single-stranded RNA (~9.6 kb) that encodes a single polyprotein. This polyprotein is cleaved into structural proteins (core, E1, E2) and non-structural (NS) proteins (NS3, NS4A, NS4B, NS5A, and NS5B) by host and viral proteases. The viral NS proteins were identified as critical components for replication and virion assembly, making them ideal targets for DAAs.

2.Role of NS3/4A Protease in Viral Maturation.

NS3/4A is a serine protease that cleaves the HCV polyprotein at specific junctions, producing functional proteins required for viral replication. Inhibiting NS3/4A blocks the processing of the polyprotein, halting replication at an early stage. Protease inhibitors (e.g., Glecaprevir, Voxilaprevir) were designed to bind the NS3/4A active site, preventing substrate processing.

3.RNA-Dependent RNA Polymerase as the Engine of Replication

NS5B is an RNA-dependent RNA polymerase responsible for synthesizing the viral RNA genome. It is highly conserved across HCV genotypes. NS5B became a key target for therapeutic intervention, as inhibiting it would directly halt genome replication. Nucleoside inhibitors (e.g., Sofosbuvir) mimic natural nucleotides and are incorporated into the viral RNA, causing chain termination. Non-nucleoside inhibitors (e.g., Dasabuvir) bind allosterically to NS5B, reducing its enzymatic activity.

4.Multifunctional Role of NS5A in Viral Replication and Assembly

NS5A is a phosphoprotein involved in multiple steps of the HCV life cycle, including replication, assembly, and interactions with host factors. NS5A inhibitors disrupt the replication complex and virion assembly, resulting in a reduction in infectious virus production. Drugs like Ledipasvir, Velpatasvir, and Pibrentasvir were designed to bind NS5A, leveraging its role as a central hub in viral replication.

5.Role of Host–Virus Interactions

HCV relies on host factors, such as lipid metabolism pathways and cellular chaperones, to facilitate its replication and assembly. Early therapies targeted host factors (e.g., interferons), but the high specificity of DAAs for viral proteins minimized off-target effects and adverse reactions. The focus on direct viral targets reduced dependency on host-directed strategies, improving efficacy and tolerability.

6.Genotype-Specific Variations.

HCV exhibits genetic diversity with seven major genotypes and numerous subtypes, which affect drug sensitivity and resistance patterns. Understanding genotype-specific differences informed the design of pan-genotypic DAAs, such as Sofosbuvir and Velpatasvir, which target conserved regions of the virus.

7.Viral Resistance Mechanisms.

HCV’s high replication rate and lack of proofreading by NS5B polymerase lead to the rapid generation of quasispecies, promoting resistance. Combination therapy targeting multiple viral proteins simultaneously was developed to prevent resistance and enhance treatment efficacy.

8.Advances in Structural Biology and Drug Design

High-resolution crystal structures of HCV proteins (e.g., NS3/4A, NS5A, and NS5B) provided a detailed understanding of their active sites and interaction dynamics. Structure-based drug design enabled the development of DAAs with high specificity and potency [66,67,68,69,70,71,72,73,74,75,76,77].

These insights into the molecular and structural biology of HCV, combined with advances in medicinal chemistry, allowed the development of DAAs that transformed HCV therapy. The ability to target multiple stages of the virus life cycle, combined with the advent of pan-genotypic agents, has resulted in cure rates exceeding 95% for most patients.

## 18. Efficacy and Improved Tolerability in IFN-Free Regimens

Clinical trials revealed that sofosbuvir, combined with ribavirin or other DAAs, achieved high SVR rates across multiple HCV genotypes. For instance, in genotype 1 patients—a group previously challenging to treat—sofosbuvir-based regimens demonstrated SVR rates exceeding 90% when combined with other DAAs such as ledipasvir [78]. Genotype 2 and 3 patients also showed excellent responses, with SVR rates around 93–97% when sofosbuvir was combined with ribavirin alone, eliminating the need for IFN [79]. This shift allowed for shorter, simpler treatment courses with fewer side effects, increasing patient adherence and satisfaction.

## 19. Impact on Treatment Adherence and Sustained Virological Response (SVR)

The shift from IFN-based to IFN-free regimens profoundly affected treatment adherence and SVR rates. IFN has been associated with a range of side effects, including flu-like symptoms, depression, and hematologic abnormalities, often leading to early treatment discontinuation. Sofosbuvir-based IFN-free regimens addressed this issue by providing a more tolerable therapy with fewer adverse events, enabling patients to complete their treatment courses. The high efficacy and low side effect profile of sofosbuvir made it feasible to offer treatment to a wider range of patients, including those previously ineligible for IFN-based regimens, such as those with advanced liver disease or liver transplantation [80,81,82,83,84].

## 20. Efficacy, Tolerability, and Reduced Treatment Duration of Combination Therapies

The combination therapies in second- and third-generation DAAs offer superior efficacy, with SVR rates often surpassing 95% and, in some cases, approaching 100%, particularly in treatment-naïve patients (Table 4). The ledipasvir–sofosbuvir combination, for example, showed exceptional efficacy for HCV genotype 1 patients, achieving SVR rates of 94–100% depending on prior treatment history and the presence of cirrhosis. This combination also demonstrated remarkable tolerability, with mild side effects such as headache and fatigue being the most reported, significantly improving the severe adverse effects of interferon-based therapies [85,86,87,88].

Another potent combination, daclatasvir–asunaprevir, has also proven effective across genotypes. Studies show that these combinations maintain high efficacy and tolerability, particularly for genotypes 1 and 3, which were historically challenging to treat. These regimens improve patient comfort and quality of life and enhance adherence, as patients are less likely to discontinue due to adverse effects [89,90].

One of the most transformative aspects of modern DAA therapies is the reduced treatment duration. Earlier regimens required 24–48 weeks of treatment, whereas modern DAA combinations can achieve high SVR rates in as little as 8–12 weeks for most patients. This shorter duration is especially beneficial for patients with complex medical profiles or limited ability to adhere to long treatment courses.

Finally, DAA therapies are also highly effective in populations previously considered difficult to treat, including patients with cirrhosis or liver transplant, those co-infected with HIV, and individuals who have relapsed after prior treatment attempts [91,92].

## 21. Pan-Genotypic DAAs: The Latest Generation

The development of pan-genotypic DAAs marks a significant advance in HCV therapy by introducing regimens effective across all major HCV genotypes. Unlike earlier treatments that required genotype-specific regimens, pan-genotypic DAAs, such as glecaprevir-pibrentasvir [93,94] and sofosbuvir-velpatasvir [95,96,97], are designed to target multiple genotypes, simplifying the treatment landscape. This shift reduces the need for genotype testing before initiating therapy, enabling faster, more accessible treatment on a global scale. These drugs not only achieve high SVR rates across genotypes but also offer improved tolerability and shorter treatment durations (8 or 12 weeks), even for hard-to-treat patient populations [98].

## 22. High SVR Rates, Improved Tolerability, and Simplified Treatment Regimens

Pan-genotypic DAAs deliver high SVR rates across HCV genotypes, even among patient populations traditionally considered challenging to treat (Table 5). Clinical trials have shown that the sofosbuvir–velpatasvir combination [99,100,101] and glecaprevir–pibrentasvir combination [102,103] achieve SVR rates exceeding 95% in patients across all major genotypes, including those with cirrhosis and previous treatment failure. The combination therapy of Sofosbuvir, Velpatasvir, and Voxilaprevir represents a powerful option, particularly for patients who have previously failed treatment with other antiviral regimens [104].

These pan-genotypic regimens are associated with minimal side effects, typically limited to mild symptoms such as headache, fatigue, and nausea. This improved tolerability represents a major shift from the adverse effect profiles of older treatments, particularly interferon-based regimens, which often caused significant side effects leading to treatment discontinuation. The reduced adverse event profile of pan-genotypic DAAs contributes to better patient adherence and overall treatment success, as patients are less likely to experience side effects that could otherwise hinder their ability to complete the regimen [105,106,107,108,109,110].

## 23. Current Recommendations and Future Perspectives

Current guidelines for hepatitis C virus (HCV) treatment from major liver societies, including the American Association for the Study of Liver Diseases (AASLD) and the European Association for the Study of the Liver (EASL) emphasize the use of pan-genotypic DAAs for simplified, highly effective treatment across all HCV genotypes [9,10]. These guidelines recommend DAAs like glecaprevir-pibrentasvir and sofosbuvir-velpatasvir as first-line treatments due to their high SVR rates, minimal side effects, and shorter treatment durations, which often range from 8 to 12 weeks for most patient populations. The guidelines also suggest routine HCV screening for all adults and prioritize treatment for high-risk groups, including those with advanced liver disease and individuals co-infected with HIV.

## 24. Remaining Challenges and Future Directions

Despite advances in HCV therapy, challenges persist. Reinfection, especially among populations with higher-risk behaviors (such as injection drug users), poses a significant obstacle to long-term viral elimination. Addressing this requires effective antiviral treatments and public health measures, such as harm reduction programs and access to preventive care [111,112,113,114]. Additionally, limited accessibility to DAAs in low-resource settings hampers global efforts to eradicate HCV. High treatment costs and inadequate healthcare infrastructure in some regions continue to restrict access, making it challenging to achieve universal HCV elimination [115].

Looking ahead, the focus is on making HCV treatment even more accessible and effective. Pan-genotypic regimens and simplified dosing protocols reduce the need for extensive pre-treatment testing and make therapy more accessible in diverse clinical settings.

As part of a global elimination effort, there is growing emphasis on community-based screening, decentralizing treatment, and developing point-of-care diagnostics to identify and treat HCV infections more rapidly [116].

Finally, a major unsolved problem remains the high cost of drugs. Reducing drug costs requires a comprehensive approach that addresses economic, regulatory and market factors [117].

Policy options could include the following:The production of generic drugs by encouraging early patent expiration or negotiating compulsory licenses to allow the production of low-cost generic versions.Centralized purchasing by governments or health organizations that can adopt large-scale supply models, negotiating directly with manufacturers to secure lower prices.Market competition to promote the entry of more manufacturers by offering tax incentives or reducing regulatory barriers, increasing competition and lowering prices.Public–private partnerships to collaborate with pharmaceutical companies to develop and produce drugs, ensuring affordable prices in exchange for research support.Public funding to invest in the research and development of treatments that are not subject to the profit-driven models of private companies.Price controls to set maximum price caps for essential medicines or impose regulations on pharmaceutical companies’ profit margins.Tax exemptions and incentives to reduce taxes or duties on raw materials and finished medicines to lower production and distribution costs.

These options must be adapted to local regulatory, economic and health contexts, while balancing affordability, innovation and sustainability for health systems.

Through these strategies, and with continued support from international health organizations, there is optimism that HCV can be eliminated as a public health threat within the coming decades.

## Figures and Tables

**Table 1 viruses-17-00163-t001:** Interferon-based therapy.

Authors	Ref	Year	Genotypes	Treatment Regimens	Weeks of Therapy	Number of Patients	SVR (%)
McHutchison et al.	[33]	1998	1,2,3	IFN α-2b 3 MU tiw	24	231	6
				IFN α-2b 3 MU tiw	48	225	13
				IFN α-2b 3 MU tiw plus RBV 1000–1200 mg/day	24	228	31
				IFN α-2b 3 MU tiw plus RBV 1000–1200 mg/day	48	228	38
Poynard	[34]	1998	1,2,3,4	IFN α-2b 3 MU tiw plus RBV 1000–1200 mg/day	48	277	43
				IFN α-2b 3 MU tiw plus RBV 1000–1200 mg/day	24	277	35
				IFN α-2b 3 MU tiw plus Placebo	48	278	19
Lindsay et al.	[35]	2001	1,2,3,4	PEG-IFN α-2b 0.5 μg/kg	48	315	18
				PEG-IFN α-2b 1.0 μg/kg	48	297	25
				PEG-IFN α-2b 1.5 μg/kg	48	304	23
				IFN α-2b 3 MU tiw	48	303	12
Zeuzem et al.	[36]	2000	1,2,3,4	PEG-IFN α-2a 180 μg	48	267	39
				IFN α-2a 6 and then 3 MU tiw	48	264	19
Heathcote et al. All patients stage 3/4	[37]	2000	1,2,3,4	PEG-IFN α-2a 90 μg	48	96	15
				PEG-IFN α-2a 180 μg	48	87	30
				IFN α-2a 3 MU tiw	48	88	12
Manns et al.	[38]	2001	1,2,3,4	PEG-IFN α-2b 1.5 μg/kg plus RBV 800 mg/day	48	511	54
				PEG-IFN α-2b 1.5 μg/kg for 4 weeks then 0.5 μg/kg for 44 weeks plus RBV 1000–1200 mg/day	48	514	47
				IFN α-2b 3 MU tiw plus RBV 1000–1200 mg/day	48	505	47
Fried et al.	[39]	2002	1,2,3,4	PEG-IFN α-2a 180 μg plus RBV 1000–1200 mg/day	48	453	56
				PEG-IFN α-2a 180 μg	48	224	29
				IFN α-2b 3 MU tiw plus RBV 1000–1200 mg/day	48	444	44
Kamal et al.	[40]	2007	4	PEG-IFN α-2b 1.5 μg/kg plus RBV (10.6 mg/kg/day)	48	50	56
				PEG-IFN α-2b 1.5 μg/kg plus RBV (10.6 mg/kg/day)	24	69	86
				PEG-IFN α-2b 1.5 μg/kg plus RBV (10.6 mg/kg/day)	36	79	76
				PEG-IFN α-2b 1.5 μg/kg plus RBV (10.6 mg/kg/day)	48	160	58
McHutchison et al.	[41]	2009	1	PEG-IFN α-2b 1.5 μg/kg plus RBV 800–1400 mg/day	48	1019	40
				PEG-IFN α-2b 1.0 μg/kg plus RBV 800–1400 mg/day	48	1015	38
				PEG-IFN α-2a 180 μg plus RBV 800–1400 mg/day	48	1035	41

Notes: MU = mega units, tiw = three times a week.

**Table 2 viruses-17-00163-t002:** Therapy with first-generation direct-acting antivirals (DAAs).

Authors	Ref	Year	Genotypes	Virological Status	Treatment Regimens	Weeks of Therapy	Number of Patients	SVR (%)
McHutchison GJ et al.	[50]	2009	1	untreated	Placebo for 12 weeks plus peginterferon alfa-2a and ribavirin for 48 weeks	48	75	41
					telaprevir, peginterferon alfa-2a and ribavirin for 12 weeks	12	17	35
					telaprevir for 12 weeks plus peginterferon alfa-2a and ribavirin for 24 weeks	24	79	61
					telaprevir for 12 weeks plus peginterferon alfa-2a and ribavirin for 48 weeks.	48	79	67
Hézode C et al.	[51]		1	untreated	telaprevir, peginterferon alfa-2a and ribavirin for 12 weeks, followed by peginterferon alfa-2a and ribavirin for 12 more weeks.	24	81	69
					telaprevir, peginterferon alfa-2a and ribavirin for 12 weeks,	12	82	48
					telaprevir and peginterferon alfa-2a for 12 weeks,	12	78	36
					peginterferon alfa-2a and ribavirin for 48 weeks.	48	81	46
McHutchison JG et al.	[52]		1	previously treated	telaprevir for 12 weeks plus peginterferon alfa-2a and ribavirin for 24 weeks;	24	115	51
					telaprevir for 24 weeks plus peginterferon alfa-2a and ribavirin for 48 weeks	48	113	53
					telaprevir for 24 weeks plus peginterferon alfa-2a and ribavirin for 24 weeks	24	111	24
					peginterferon alfa-2a and ribavirin for 48 weeks	48	114	14
Jacobson IM et al.	[53]		1	untreated	telaprevir combined with peginterferon alfa-2a and ribavirin for 12 weeks (T12PR group), followed by peginterferon-ribavirin alone for 12 weeks if HCV RNA was undetectable at weeks 4 and 12 or for 36 weeks if HCV RNA was detectable at either time point;	24–36	363	75
					telaprevir with peginterferon-ribavirin for 8 weeks and placebo with peginterferon-ribavirin for 4 weeks (T8PR group), followed by 12 or 36 weeks of peginterferon-ribavirin on the basis of the same HCV RNA criteria;	36–48	364	69
					placebo with peginterferon-ribavirin for 12 weeks, followed by 36 weeks of peginterferon-ribavirin	48	361	44
Zeuzem S et al.	[54]		1	previously treated	the T12PR48 group, received telaprevir for 12 weeks and peginterferon plus ribavirin for a total of 48 weeks;	48	266	83
					T12PR48 group, received 4 weeks of peginterferon plus ribavirin followed by 12 weeks of telaprevir and peginterferon plus ribavirin for a total of 48 weeks;	48	264	88
					peginterferon plus ribavirin for 48 weeks.	48	132	24
Kwo PY et al.	[55]		1	untreated	peginterferon alfa-2b plus ribavirin for 48 weeks	48	104	38
					peginterferon alfa-2b plus ribavirin for 4 weeks, followed by peginterferon alfa-2b, ribavirin, and boceprevir for 24 weeks	28	103	54
					peginterferon alfa-2b plus ribavirin for 4 weeks followed by peginterferon alfa-2b, ribavirin, and boceprevir for 44 weeks	48	103	56
					peginterferon alfa-2b plus ribavirin and boceprevir for 28 weeks	28	107	67
					peginterferon alfa-2b plus ribavirin and boceprevir for 48 weeks.	48	103	75
Poordad F et al.	[56]		1	untreated	peginterferon alfa-2b and ribavirin were administered for 4 weeks (the lead-in period) and subsequently, placebo plus peginterferon-ribavirin for 44 weeks;	48	363	38
					peginterferon alfa-2b and ribavirin were administered for 4 weeks (the lead-in period) and subsequently boceprevir plus peginterferon-ribavirin for 24 weeks, and those with a detectable HCV RNA level between weeks 8 and 24 received placebo plus peginterferon-ribavirin for an additional 20 weeks	48	368	63
					peginterferon alfa-2b and ribavirin for 4 weeks (the lead-in period) and subsequently boceprevir plus peginterferon-ribavirin for 44 weeks.	48	366	66
Bacon BR et al.	[57]		1	previously treated	peginterferon alfa-2b and ribavirin for 4 weeks (the lead-in period) and subsequently placebo plus peginterferon-ribavirin for 44 weeks;	48	80	21
					peginterferon alfa-2b and ribavirin for 4 weeks (the lead-in period) and subsequently boceprevir plus peginterferon-ribavirin for 32 weeks, and patients with a detectable HCV RNA level at week 8 received placebo plus peginterferon-ribavirin for an additional 12 weeks;	48	162	59
					peginterferon alfa-2b and ribavirin were administered for 4 weeks (the lead-in period) and subsequently received boceprevir plus peginterferon-ribavirin for 44 weeks.	48	161	66

**Table 3 viruses-17-00163-t003:** DAAs of the second and third generation with the protein inhibited, typical combination and year of approval.

DAA	Protein Inhibited	Typical Combination	Year of Approval
**Sofosbuvir**	NS5B (polymerase)	Sofosbuvir + Ledipasvir	2013
**Ledipasvir**	NS5A	Sofosbuvir + Ledipasvir	2014
**Ombitasvir**	NS5A	Ombitasvir + Paritaprevir + Dasabuvir	2014
**Paritaprevir**	NS3/4A (protease)	Ombitasvir + Paritaprevir + Dasabuvir	2014
**Dasabuvir**	NS5B (non-nucleoside)	Ombitasvir + Paritaprevir + Dasabuvir	2014
**Asunaprevir**	NS3/4A (protease)	Beclabuvir + Daclatasvir + Asunaprevir	2014
**Daclatasvir**	NS5A	Sofosbuvir + Daclatasvir	2015
**Beclabuvir**	NS5B (non-nucleoside)	Beclabuvir + Daclatasvir + Asunaprevir	2015
**Velpatasvir**	NS5A	Sofosbuvir + Velpatasvir	2016
**Elbasvir**	NS5A	Grazoprevir + Elbasvir	2016
**Grazoprevir**	NS3/4A (protease)	Grazoprevir + Elbasvir	2016
**Glecaprevir**	NS3/4A (protease)	Glecaprevir + Pibrentasvir	2017
**Pibrentasvir**	NS5A	Glecaprevir + Pibrentasvir	2017
**Voxilaprevir**	NS3/4A (protease)	Sofosbuvir + Velpatasvir + Voxilaprevir	2017

**Table 4 viruses-17-00163-t004:** Therapy with second-generation direct-acting antivirals (DAAs).

Authors	Ref	Year	Genotypes	Virological Status	Treatment Regimens	Weeks of Therapy	Number of Patients	SVR (%)
Afdhal N et al.	[84]	2014	1	untreated	sofosbuvir-ledipasvir	12	214	99
					sofosbuvir-ledipasvir plus ribavirin	12	217	97
					sofosbuvir-ledipasvir	24	217	98
					sofosbuvir-ledipasvir plus ribavirin	24	217	99
Afdhal N et al.	[85]	2014	1	previously treated	sofosbuvir-ledipasvir	12	109	94
					sofosbuvir-ledipasvir plus ribavirin	12	111	96
					sofosbuvir-ledipasvir	24	109	99
					sofosbuvir-ledipasvir plus ribavirin	24	111	99
Lawitz E et al.	[86]	2014	1	untreated (Cohort A)	sofosbuvir-ledipasvir	8	20	95
					sofosbuvir-ledipasvir plus ribavirin	8	21	100
					sofosbuvir-ledipasvir	12	19	95
			1	previously treated with a protease-inhibitor regimen (Cohort B)	sofosbuvir-ledipasvir	12	19	95
					sofosbuvir-ledipasvir plus ribavirin	12	20	100
Kowdley KV et al.	[87]	2014	1	untreated	sofosbuvir-ledipasvir	8	215	94
					sofosbuvir-ledipasvir plus ribavirin	8	216	93
					sofosbuvir-ledipasvir	12	215	95
Poordard F et al.	[88]	2015	1	untreated	daclatasvir—asunaprevir beclabuvir	12	312	92
				previously treated	daclatasvir—asunaprevir-beclabuvir	12	103	89
Muir AJ et al.	[89]	2015	1	untreated cirrhosis	daclatasvir—asunaprevir-beclabuvir	12	57	93
					daclatasvir—asunaprevir-beclabuvir plus ribavirin	12	55	98
				previously treated cirrhosis	daclatasvir—asunaprevir-beclabuvir	12	45	86
					daclatasvir—asunaprevir-beclabuvir plus ribavirin	12	45	93

**Table 5 viruses-17-00163-t005:** Therapy with pan-genotypic direct-acting antivirals (DAAs).

Authors	Ref	Year	Genotypes	Virological Status	Treatment Regimens	Weeks of Therapy	Number of Patients	SVR (%)
Feld JJ et al.	[99]	2015	1,2,4,5,6	untreated (68%) previously treated (32%)	Sofosbuvir–Velpatasvir	12	624	99
					Placebo	12	116	0
Foster GR et al.	[100]	2015	2	untreated previously treated	Sofosbuvir–Velpatasvir	12	134	99
					Sofosbuvir–Ribavirin	12	132	94
			3	untreated previously treated	Sofosbuvir–Velpatasvir	12	277	95
					Sofosbuvir–Ribavirin	24	275	80
Curry MP et al.	[101]	2015	1,2,3,4,5,6	untreated and previously treated decompensated cirrhosis	Sofosbuvir–Velpatasvir	12	90	83
					Sofosbuvir–Velpatasvir plus Ribavirin	12	98	94
					Sofosbuvir–Velpatasvir	24	92	86
Forns X et al.	[102]	2017	1,2,4,5,6	untreated and previously treated compensated cirrhosis	Glecaprevir-Pibrentasvir	12	146	99
Zeuzem S et al.	[103]		1	untreated previously treated	Glecaprevir-Pibrentasvir	8	351	99
					Glecaprevir-Pibrentasvir	12	352	99
			3	untreated	Glecaprevir-Pibrentasvir	12	233	95
					Sofosbuvir-Daclatasvir	12	115	97
					Glecaprevir-Pibrentasvir	8	157	95
Bourlière M et al.	[104]		1 (Polaris-1)	previously treated with a regimen containing an NS5A inhibitor	Sofosbuvir–Velpatasvir–Voxilaprevir	12	263	96
					Placebo	12	152	0
			1,2,3 (Polaris-4)	previously treated without a regimen containing an NS5A inhibitor	Sofosbuvir–Velpatasvir–Voxilaprevir	12	182	98
					Sofosbuvir–Velpatasvir	12	151	90

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
