# Peer review of "A Comprehensive Review of Antiviral Therapy for Hepatitis C: The Long Journey from Interferon to Pan-Genotypic Direct-Acting Antivirals (DAAs)"

_viruses, 2025, doi:10.3390/v17020163_

Round 1
Reviewer 1 Report
Comments and Suggestions for Authors
This is a fine narrative review of the stages of development of HCV therapy. It reads well and is clear, but the purpose of the review could be better refined, and the information provided targeted to aid the reader.
Personally I would find it more engaging if included less data about the efficacy of the older regimens (which is available elsewhere and of only historical interest) and included more of the motivations for development of better HCV therapy (and the key underlying biologic insights that led to development of these drugs). It would also benefit from discussion of therapies or potential approaches that failed - approaches that were once considered promising that didn't work out. Also consider including the implications of the treatment developments, in terms of the scalability and accessibility of treatment, and their public health implications.
Overall I think refinement of the purpose and including some of these additional aspects would enhance interest for the reader.
Author Response
Comment 1 :
This is a fine narrative review of the stages of development of HCV therapy. It reads well and is clear, but the purpose of the review could be better refined, and the information provided targeted to aid the reader.
Response 1: We have redefined the purpose and aims of the article in the introductory section.
Personally I would find it more engaging if included less data about the efficacy of the older regimens (which is available elsewhere and of only historical interest) and included more of the motivations for development of better HCV therapy (and the key underlying biologic insights that led to development of these drugs). It would also benefit from discussion of therapies or potential approaches that failed - approaches that were once considered promising that didn't work out. Also consider including the implications of the treatment developments, in terms of the scalability and accessibility of treatment, and their public health implications.
Response 1 :
We responded to the review comment by restructuring the section dedicated to DAAs. We included in the article a section entitled "Biological insights that led to development of DAAs" that describes the steps of discovery and the biological basis of new drugs. Furthermore, at the end of the article we reported some considerations on the accessibility of DAAs with a clear reference to a perspective article published in the New England Journal of Medicine on January 1, 2025 ( Reference 117: Tu S.S. Shyamasundaran Kottilil , Mattingly T. J. Leveraging Old Hepatitis C Therapies N Engl J Med. 2025; 392:2-4)
Reviewer 2 Report
Comments and Suggestions for Authors
Dear Authors.
The review manuscript entitled “A comprehensive review of antiviral therapy for hepatitis C: The long journey from Interferon to Pangenotypic direct-acting antivirals (DAAs)” concisely summarises the advances of HCV therapy through the years. In most parts, the manuscript is clearly written and well organised. A few minor points listed below may help improve the manuscript:
Specific comments:
1. Introduction + reference 1, the newest WHO fact sheet estimates 50 million people chronically infected and 1 million new infections.
2. Page 2 Typo in “clinical efficacy and challenges” Unfortunately, relapse “was” common…
3. Page 2 “understanding the limitations”, maybe here or in the interferon-free regimen, it could be highlighted the disadvantages of the subcutaneous injection of IFN vs oral administration of the DAAs
4. Page 3/4-“Mechanism of action of Ribavirin…”, this paragraph would benefit further revision, For example, the “not fully understood could be removed as there are several articles/reviews that go into the study of ribavirin action. One mode of action, IMPDH is not mentioned and more detail on the inhibition of RNA replication would be beneficial (i.e. Ribavirin triphosphate blocks HCV RNA elongation by incorporating into the RNA opposite a uridine or cytidine. The review article, “Mechanism of Action of Ribavirin in the Treatment of Chronic Hepatitis” by Te et al 2007 Gastroenterol Hepatol nicely explains the modes of action of ribavirin.
5. Table 1, a small footnote that explains what is MU and tiw is required for clarity.
6. Page 9, second generation DAAs, it would be useful if all the available 2nd and third-generation DAAs were listed in a table or explained in more detail in the text together with which protein they inhibit and which DAAs they are normally combined, rather than just writing “other DAAs”.
Author Response
Comment 1: Introduction + reference 1, the newest WHO fact sheet estimates 50 million people chronically infected and 1 million new infections.
Response: we have corrected the data in the text
Comment 2: Page 2 Typo in “clinical efficacy and challenges” Unfortunately, relapse “was” common…
Response : we have correct the mistake
Comment 3: Page 2 “understanding the limitations”, maybe here or in the interferon-free regimen, it could be highlighted the disadvantages of the subcutaneous injection of IFN vs oral administration of the DAAs
Response: We reviewed the sentence and corrected it with the suggestions indicated by the review
Comment 4: Page 3/4-“Mechanism of action of Ribavirin…”, this paragraph would benefit further revision, For example, the “not fully understood could be removed as there are several articles/reviews that go into the study of ribavirin action. One mode of action, IMPDH is not mentioned and more detail on the inhibition of RNA replication would be beneficial (i.e. Ribavirin triphosphate blocks HCV RNA elongation by incorporating into the RNA opposite a uridine or cytidine. The review article, “Mechanism of Action of Ribavirin in the Treatment of Chronic Hepatitis” by Te et al 2007 Gastroenterol Hepatol nicely explains the modes of action of ribavirin.
Response: We have revised the paragraph on the action of ribavirin and remodeled it with the reviewer's advice. We have also added the indicated review as reference 32
Comment 5: Table 1, a small footnote that explains what is MU and tiw is required for clarity.
Response: the footnote has been added to table 1
Comment 6: Page 9, second generation DAAs, it would be useful if all the available 2nd and third-generation DAAs were listed in a table or explained in more detail in the text together with which protein they inhibit and which DAAs they are normally combined, rather than just writing “other DAAs”.
Response: The section on the discovery of DAAs and their biological action has been revised as suggested by reviewer 1 and a table with targets, therapeutic combinations and year of approval has been added as suggested by reviewer 2.